# Proliferation and Cluster Analysis of Neurons and Glial Cell Organization on Nanocolumnar TiN Substrates

**DOI:** 10.3390/ijms21176249

**Published:** 2020-08-28

**Authors:** Alice Abend, Chelsie Steele, Sabine Schmidt, Ronny Frank, Heinz-Georg Jahnke, Mareike Zink

**Affiliations:** 1Soft Matter Physics Division and Biotechnology & Biomedical Group, Peter-Debye-Institute for Soft Matter Physics, Leipzig University, Linnéstr. 5, 04103 Leipzig, Germany; alice.abend@uni-leipzig.de (A.A.); cs73gygu@studserv.uni-leipzig.de (C.S.); 2Centre for Biotechnology and Biomedicine, Molecular Biological-Biochemical Processing Technology, Leipzig University, Deutscher Platz 5, 04103 Leipzig, Germany; sabine.schmidt@bbz.uni-leipzig.de (S.S.); ronny.frank@bbz.uni-leipzig.de (R.F.)

**Keywords:** neurons, glial cells, electrode materials, autocorrelation function, cluster analysis, cell proliferation, TiN, nanocolumnar surface

## Abstract

Biomaterials employed for neural stimulation, as well as brain/machine interfaces, offer great perspectives to combat neurodegenerative diseases, while application of lab-on-a-chip devices such as multielectrode arrays is a promising alternative to assess neural function in vitro. For bioelectronic monitoring, nanostructured microelectrodes are required, which exhibit an increased surface area where the detection sensitivity is not reduced by the self-impedance of the electrode. In our study, we investigated the interaction of neurons (SH-SY5Y) and glial cells (U-87 MG) with nanocolumnar titanium nitride (TiN) electrode materials in comparison to TiN with larger surface grains, gold, and indium tin oxide (ITO) substrates. Glial cells showed an enhanced proliferation on TiN materials; however, these cells spread evenly distributed over all the substrate surfaces. By contrast, neurons proliferated fastest on nanocolumnar TiN and formed large cell agglomerations. We implemented a radial autocorrelation function of cellular positions combined with various clustering algorithms. These combined analyses allowed us to quantify the largest cluster on nanocolumnar TiN; however, on ITO and gold, neurons spread more homogeneously across the substrates. As SH-SY5Y cells tend to grow in clusters under physiologic conditions, our study proves nanocolumnar TiN as a potential bioactive material candidate for the application of microelectrodes in contact with neurons. To this end, the employed K-means clustering algorithm together with radial autocorrelation analysis is a valuable tool to quantify cell-surface interaction and cell organization to evaluate biomaterials’ performance in vitro.

## 1. Introduction

The human brain is such a complex system that its composition and architecture are still not fully understood. Even the number of neurons and glial cells in the brain remains questionable [1,2]. Besides structural heterogeneities within the brain and related unsolved questions in neuronal science [3], in light of the currently increasing numbers of cases of neurodegenerative diseases such as Parkinson’s disease, the study of cell behavior and cellular function in the brain is more important than ever before [4].

In vivo animal studies of diseases, such as Parkinson’s disease, are difficult to assess because they comprise a varying age of onset, symptoms, and rate of progression. This heterogeneity requires the use of a variety of animal models to study different aspects of the disease [5]. Alternatively, organotypic cultures offer the possibility to investigate brain tissue slices ex vivo [6]. However, especially for adult mammalian tissues, organotypic preservation is difficult and tissue distortion often takes place within a few days in culture. As shown by Kallendrusch et al., nanostructured surfaces employed as tissue scaffolds, such as titanium dioxide nanotube arrays, can overcome this issue and allow us to culture adult tissue slices of the brain for at least 10 days [7].

In vitro cell cultures are much easier systems to use to assess cellular function/dysfunction and the effect of drugs to develop new treatments and tailored therapies [8,9]. Moreover, cell cultures also offer good testbeds to study the interaction of neurons and other brain cells in contact with biomaterials employed, e.g., for application as brain pacemaker devices for deep brain stimulation [10]. The interaction of the neurons with the surface of the pacemaker’s electrodes plays a major role in the functionality of the device and, consequently, therapy success [11]. Thus, research focuses on the fabrication of biocompatible materials that promote cell adhesion, proliferation, and physiological function and provide stable charge transfer at the brain/machine interface [12]. These materials are, for example, based on metals [13], carbon [14,15], or silicone compounds [16]. Commonly used electrode materials for the electrochemical analysis of biological samples, such as cells and tissues, are noble metals, like gold and platinum, because of their high conductivity, chemical stability, and biocompatibility [17,18,19,20,21].

Not only the material itself but also the surface topography plays an important role in the interaction with cells [22]. Surface topography designs vary from simple microgrooves [23,24] and micrometer-sized pillars [25,26] down to nanofabricated structures [27], nanowires [28,29], nanopillars [30], and nanotubes [31], and can also be combined with novel surface coatings [32,33].

Beyond that, there are alternative electrode materials that offer application-specific advantages, like optical transparency (indium tin oxide, ITO [17]) or an increased surface area (titanium nitride, TiN [34]). The latter allows the shrinking of microelectrode size without losing detection sensitivity due to a lowered self-impedance of the electrode [35,36,37]. These materials offer great perspectives for in vitro lab-on-a-chip devices, such as multielectrode arrays (MEA). For example, as shown by Jahnke et al., MEAs have already been successfully employed for in vitro screenings of hallmarks of neurodegenerative diseases by impedance spectroscopy [38]. Additionally, microelectromechanical systems (MEMS) and microsystems composed of the above-mentioned materials have enabled the study of neurons from the single unit to the scale of large populations and neural circuits (for an overview, see Ref. [39,40]). Lab-on-a-chip formats even allow the combination of electrical function with optical analysis and biochemical patterning to enhance cell-surface interaction [41].

For any in vitro cell cultures with the aim to determine the interaction of biomaterials with cells, it is of great importance to define quantitative measures that determine if a biomaterial is bioactive and supports proliferation on the surface. Research methods to examine the compatibility of the electrode material often involves immunostaining of relevant cellular components. In addition, the number of cells and cell division rates can easily be determined. However, how cells organize on the surface, if they homogeneously spread or agglomerate, is often neglected, and a quantitative measure of cellular organization is missing. For example, under physiologic conditions, neurons such as SH-SY5Y—an established human neuroblastoma cell line to study Parkinson’s disease—tend to cluster on a surface [42].

Radial autocorrelation functions and cluster formation algorithms can offer powerful tools to quantify the spatial organization of particles up to cells on two-dimensional surfaces, as well as three-dimensional environments. While radial autocorrelation functions have been employed before to determine spatial correlations of particles in supercooled liquids [43], Pan et al. [44] showed that autocorrelation functions calculated for cultured cells can also be used to quantify cell sizes.

Here, we show how proliferation assays, in combination with a quantitative cellular organization analysis performed by radial autocorrelation functions and clustering analysis, can be used to quantify cellular performance on potential biomaterials. In our study, we investigated the behavior of neurons and glial cells on different surfaces such as TiN substrates and nanocolumnar TiN, which exhibit an increased surface area compared to TiN with larger grain sizes. The interaction of cells with this nanocolumnar TiN has never been studied before, and good biocompatibility would offer great potential for the development of miniaturized multielectrode arrays as described above. Our results show a clear superiority of these materials in terms of cell division rate and the cellular organization of neurons in contrast to cell behavior on gold and ITO surfaces. The latter two materials were employed for control experiments as these materials are considered non-toxic and are often the materials of choice for electrodes in contact with neurons [45]. Future applications of the presented nanocolumnar TiN materials aside, our combined analysis tool of clustering algorithms and radial autocorrelation calculations allows for a fast evaluation of biomaterials’ performance in vitro, by simply measuring cell positions from fluorescent images.

## 2. Results

### 2.1. Topographies of Electrode Materials

Before investigating the interaction of neurons and glial cells with different electrode materials, the topographies of the surfaces were characterized by atomic force microscopy (AFM), as shown in Figure 1. The gold substrates exhibited the smoothest surface features with a root-mean-square (RMS) roughness of (2.95 ± 1.63) nm, comparable to the thin TiN coating with (2.98 ± 1.24) nm. By contrast, ITO showed the highest RMS roughness of the tested materials with a value of (8.36 ± 0.99) nm, also significantly exceeding the thick TiN layers (in the following, termed TiN nano and nanocolumnar TiN) with a RMS roughness of (6.42 ± 0.99) nm. With respect to the surface area increase (viz. the dimensionless ratio of surface area to projected area), TiN nano showed the highest increase with 1.27 ± 0.08 of the projected surface area, while the other materials were below 1.1 (Au: 1.02 ± 0.01, ITO: 1.10 ± 0.02, TiN: 1.07 ± 0.01).

Besides the different surface roughness, varying-grain-sizes of the different surfaces became visible (see Figure 1). While Au exhibited smooth transitions between the grains with a mean grain size of (82 ± 10) nm, ITO showed clearly distinguishable crystallites with a larger mean grain size of (109 ± 19) nm. Besides different film thicknesses of the TiN layers due to different sputter times: 150–200 nm for TiN and 500–550 nm for TiN nano, their surface morphologies differed remarkably. While TiN exhibited a cauliflower motif with a mean grain size of (90 ± 11) nm and subgrains of (17 ± 4) nm, TiN nano appeared to have a nanocolumnar structure with sharply delimited single-type grains with a size of (38 ± 9) nm, being the origin of the high surface area increase.

### 2.2. Cell Growth on Electrode Materials

In order to investigate neuronal and glial cell behavior on potential electrode materials, the human neuroblastoma cell line SH-SY5Y and the human glioblastoma cell line U-87 MG were grown on the four different electrode materials presented above. Cells were fluorescently labeled, imaged, and subsequently counted one and three days after seeding for the glial cell type, while the number of neuronal cells was investigated 1 and 3 days after differentiation. The results of the average cell numbers for each substrate are shown in Figure 2.

For the neuronal cells, within the first day after differentiation, the number of cells on all four substrates shows no statistical difference. Around 2000 cells adhered to all surfaces. However, after 3 days on ITO, the cell number remained constant and even halved on Au, while on TiN and TiN nanocolumnar surfaces, cells proliferated with an around three-fold increase to approximately 5400 cells on TiN and 6000 cells on nanocolumnar TiN.

Similar results were found for the glial cells: 1 day after seeding, similar cell numbers were seen for Au (2400 cells), TiN (2600 cells), and TiN nanocolumnar substrates (2700 cells) and fewer cells on ITO (1800 cells). Two days later, cell numbers more than doubled to approximately 6000 cells with Au as the only outlier on which we counted approximately 4000 cells, thus 2000 cells less than on the other materials.

Comparing the experimental results for the neuronal SH-SY5Y and glial U-87 MG cells, we observed a similar growth behavior on TiN and TiN nanocolumnar substrates for both cell types. Here, seeding the same number of cells led to equal numbers of cells for short and longer culture times. The situation for gold and ITO materials seems to be completely different. The SH-SY5Y cells did not proliferate as fast on these materials as the U-87 MG cells. We found about three times more U-87 MG cells on ITO substrates as SH-SY5Y cells for the longer growth time. For the gold material, that factor rose to four, while the SH-SY5Y cell population decreased, and the U-87 MG cell number grew.

### 2.3. Radial Autocorrelation of Cell Positions

We performed a radially averaged autocorrelation analysis for the cell nuclei positions for all 48 samples, viz. glia and neuronal cells cultured on Au, ITO, TiN, and TiN nanocolumnar substrates for 1 and 3 days. Representative results of the radially averaged autocorrelation functions are presented in Figure 3. As shown by Baker et al. [46], the undulating autocorrelation curves represent a homogeneous distribution of objects and uniform object size. The first minimum of the curves marks the typical size of objects in an image, whereas the first peak gives an estimate on the object spacing. On the other hand, flattened autocorrelation curves indicate an inhomogeneous distribution of objects and several different object sizes in an image. The point where the curve bends from a steep slope to an almost constant regime characterizes the average size of objects.

We see rapidly decreasing autocorrelation functions for the U-87 MG cells on all materials (green curves in Figure 3). The graphs of the 1 day culture experiments show a small but noticeable undulating form with distinct first minima (20 µm for gold and ITO, 50 µm for TiN materials). Additionally, for gold, the curve displays a peak at about 30 µm and for ITO and both TiN materials, around 70–80 µm, which mark the distance to the next object. Comparing the graphs to the original fluorescent images, we notice an even distribution of U-87 MG cells, viz. the cell population was homogeneously spread over the entire surface. In fact, the observed correlation length of approximately 20 µm for the U-87 MG cells cultured on gold for 1 day corresponds to objects composed of two cell nuclei in the fluorescent image, while single-cell nuclei were also present on the substrate. By contrast, the radially averaged autocorrelation curves of glial cells cultured for 3 days on gold do not show the undulations anymore. Looking at the associated cell images revealed that the cells still grew uniformly distributed over the entire substrate area. Nevertheless, small cell agglomerations of various sizes became visible. We see objects of about 20 to 40 µm in diameter, which corresponds to agglomerations of 2 to 5 cells in the images of glial cells cultured on gold substrates for 3 days. Thus, U-87 MG cells proliferated rapidly, as can be seen from Figure 2, and grew homogeneously distributed on all tested materials. Such behavior became visible in the autocorrelation curves after 3 days of culture, which are shifted toward larger distances in comparison to their 1 day counterparts, indicating the existence of (on average) larger cell agglomerations. A very similar behavior of glial cells became present on the ITO substrates. After one day of culture, mainly single cells and pairs were homogeneously distributed on the surface, reflected by the autocorrelation curve minimum around 20 μm, while, after 3 days, smaller aggregates—still homogeneously distributed—were seen, represented by a shift in the autocorrelation minimum toward larger correlation distances around 40–50 μm. Additionally, we observed larger cell agglomerations (50 µm) on TiN and TiN nanocolumnar substrates in comparison to their gold and ITO counterparts after 1 day of culture. These objects are still homogeneously distributed. The correlation length is shifted for TiN and nanocolumnar TiN after additional growth time, and the comparison with fluorescent images reveals cell agglomerations of various sizes up to about 100 µm.

However, in the case of the SH-SY5Y cells (red curves in Figure 3), we see considerably different results. While, after 1 day, the correlation length, viz. the first minimum where the steep slope transits into an almost constant regime, was found for 60 μm (Au), 70 μm (ITO), 90 μm (TiN), and 160 µm (nanocolumnar TiN), there are noticeable shifts in the correlation curves to longer distances for longer culture times on all substrate types. This long-range correlation indicates the growth of cell clusters of various sizes. We do not see any undulating graphs for the gold and ITO substrates. Comparison with fluorescent images revealed cell agglomerations of various sizes, which correlate with the position of the minimum of the autocorrelation graphs. However, for the neuronal cells grown on TiN and nanocolumnar TiN for 1 day, the autocorrelation curves, in fact, show the undulating form as similarly seen for glial cells on Au. However, this behavior vanished after additional culture time, and we noticed correlation lengths of up to 500 µm for TiN samples and even larger values for nanocolumnar TiN. Figure 4 shows the autocorrelation curves for all experiments of SH-SY5Y cells with three days of culture time. The three graphs with the prominent steep decrease correspond to experiments where the cells formed especially large agglomerations of about 1200–2000 µm. Overall, we see cluster formation of greatly varying sizes on different substrate types but also between individual experiments on the same material.

### 2.4. Cell Clustering Results

To further investigate the spatial distribution and the formation of cellular clusters on the substrates, we employed a K-means algorithm as described in the Methods. As we saw that the glial cells were almost evenly distributed over the surface of all materials, we only considered the neuronal cell distributions here. One example of such an analysis is shown in Figure 5. Each data point represents a single SH-SY5Y cell grown on a TiN nanocolumnar surface—corresponding to the autocorrelation graph marked with a * in Figure 4. The algorithm sorted the cells into four clusters, indicated by the different colors. Although there seems to be an overlap of clusters, every cell belongs to exactly one cluster and is not counted twice. The ellipses mark the area of the clusters surrounding the cluster centroid, while there are only a few outliers visible. In order to validate if the calculated cluster number is correct, we compared the results by employing the elbow method and gap statistics as described in the Materials and Methods. The corresponding elbow graph denotes the optimal cluster number at the point where the steep decline bends over to a flattened regime. Here, we found this bending point of the elbow graph (Appendix A) for an ideal cluster number of four for the example of neuronal cells on a TiN nanocolumnar substrate—in line with the results from the K-means algorithm. Moreover, our results from gap statistics (see Appendix A) corroborate four as the ideal cluster number by showing a maximum peak at k = 4.

Additionally, we used the silhouette method (see Materials and Methods) to verify our choice of the optimal number of clusters for each experiment and validate if the cells are sorted into the right cluster. Appendix A shows the average silhouette coefficient as a function of cluster numbers. The peak of the curve marks the optimal number of clusters. In this example, it is four, which corroborates our choice of four clusters. If the silhouette width turned out to be lower than a certain threshold, which was chosen to be 0.35 after several test simulations, the cluster number was considered wrong. Thus, other cluster numbers were iteratively tested until the silhouette width exceeded the value of 0.35. Figure 6 shows the silhouette plot for neuronal cells on nanocolumnar surfaces corresponding to the cluster analysis of Figure 5 (here, the calculated silhouette width was 0.48). This indicates a high-quality clustering result as there are very few falsely grouped cells, which would be indicated by negative silhouette coefficient values.

Besides the cluster analysis shown above for neuronal cells on TiN nanocolumnar surfaces after 3 days, we used the K-means clustering algorithm to investigate cluster formation on all substrate materials. As shown in Figure 7, we can see a great difference in the clustering behavior of SH-SY5Y cells: The density of the cell clusters is much lower on gold and ITO substrates than on TiN-containing substrates. The clusters do not grow significantly denser on gold and ITO with longer cell growth times. By contrast, the clusters of cells grown on TiN and TiN nanocolumnar substrates double their density on average for the longer growth time.

## 3. Discussion

In our study, we investigated the proliferation and organization of neuronal (SH-SY5Y) and glial (U-87 MG) cells on planar and nanocolumnar TiN, which offer great potential for application as a multielectrode array material. Results were compared to cell behavior on ITO and gold surfaces as both materials are well-known to promote adhesion of neurons and are used for neural stimulation systems and brain/machine interfaces [47,48]. Furthermore, by employing radial autocorrelation functions in combination with clustering analysis, we quantified the cellular arrangement on the surfaces. We found that high cell numbers, viz. fast proliferation, do not necessarily lead to large cell clusters. In the case of U-87 MG cells, we obtained rapid cell proliferation on all materials, while only the formation of small cell agglomerations in comparison to their neuronal counterparts was seen. Judging from the corresponding radially averaged autocorrelation functions and fluorescent images, the glial cells form small cell agglomerations of different sizes rather than large clusters after 3 days of culture time. However, SH-SY5Y cells form large clusters on TiN and nanocolumnar TiN substrates after 3 days of growth, and we saw the highest neuronal cell numbers on these materials. Surprisingly, the SH-SY5Y cell numbers remained constant on ITO and even shrank on gold substrates for longer growth times, while the cellular organization changed, and agglomerations of different sizes became visible as similarly observed for U-87 MG cells. Nevertheless, glial cells proliferated much faster under the same conditions on these materials. Even though gold and ITO are considered non-toxic as mentioned before, the reduced proliferation rate of SH-SY5Y cells points toward altered physiology and metabolism, which should be addressed in future studies.

For a more detailed analysis of neuronal cluster formation, we employed K-means cluster analysis of the SH-SY5Y cell experiments. Here, the cell density inside clusters mirrored the cell proliferation behavior of the neuronal cells on planar TiN and TiN nanocolumnar substrates. Thus, higher overall cell numbers resulted in denser cell clusters for these materials. On the other hand—although not statistically significant—the cell density in clusters grown on gold and ITO increased with growth time, while the overall cell number on these substrates decreased in the case of gold (statistically significant) and stayed constant on ITO.

SH-SY5Y cells on gold and ITO formed several small agglomerations of different sizes, which are scattered homogeneously over the substrate. On first sight, the K-means algorithm contrarily sorts these agglomerations in no more than four clusters. Nevertheless, the cluster algorithm results are consistent with the outcome of the autocorrelation curve as well as our visual inspections of the fluorescent images. Here, the cell patterns hardly changed after additional culture time, and the cells still grew homogeneously distributed but arranged in small agglomerations of different sizes. These small clusters grew over time, which is also represented by the shifted radially averaged correlation length; however, the entire cell patterns did not change fundamentally, i.e., no large and dense clusters were formed. Such behavior is indeed reflected by the K-means algorithm, which did not give any statistically significant change in cell density in clusters for the gold and ITO materials for neuronal cells with longer culture times. Thus, we conclude that the K-means clustering algorithm works well for detecting large, dense cell clusters (on TiN and nanocolumnar TiN in our case) but fails to identify smaller cell agglomerations and, instead, pools them into bigger but therefore less dense clusters. For small agglomerations, the radially averaged autocorrelation function can reliably quantify such cellular arrangements and also predicts large cell clusters. Similar observations were reported by Baker et al. by applying the radially averaged autocorrelation method to analyze natural quartz crystal patterns [46]. Very long correlation lengths of several hundred micrometers always occurred for our experiments where we saw especially large cell clusters. We cannot read the actual size of the largest object directly from the autocorrelation graphs, due to the blurring effect of the radial averaging (here, the K-means algorithm is the analysis of choice); however, the curve gives an average size of the objects on the respective substrates.

While the proliferation data clearly show that the TiN and nanocolumnar TiN substrates support cell division best, the radial autocorrelation function in combination with cluster analysis alone cannot indicate if a surface exhibits optimal conditions for the in vitro cell culture. As shown by Chan-Ling et al., as well as Ogata et al., a physiological pattern for astrocytes is a homogeneously distributed cell layer where only the most peripheral cell processes are in contact with neighboring cells [49,50]. Cells are shaped more like polyhedrons instead of stars [50]. The star-like shape stems from the distribution of fluorescently labeled glial fibrillary acidic protein (GFAP) in cells [51]. An in-depth review of the role of astrocytes in the function and architecture of the brain is available from Nedergaard et al. [52]. In our study, we can see from the associated fluorescent images of U-87 MG cells that the cells look indeed more like polyhedrons than stars. Moreover, our cells show a homogeneous distribution for short and longer culture times but develop small cell agglomerations. In this context, glial cells form physiological patterns on all investigated material types in our experiments in vitro. Thus, comparison of fluorescent images with cluster analysis is a valuable tool for improved quantification of biocompatible surfaces.

Considering the effect of surface topography, Vallejo-Giraldo et al. investigated SH-SY5Y cells grown on ITO substrates with varying surface roughness [47]. They reported that the semi-rough substrates (R_a_ = 19 nm) performed best in terms of cell growth, while neuronal cells do not attach well on very smooth (R_a_ = 1 nm) or especially rough surfaces (R_a_ = 81 nm) [47]. Thus, poor proliferation of SH-SY5Y cells on ITO in comparison to the TiN substrates might be attributed to the surface structure, which hinders cell adhesion. Khan et al. came to similar conclusions while investigating the adhesion of neuronal cells (rat cortical neurons) on silicon wafers [53]. According to their study, neuron adherence increases with substrate surface roughness until a certain limit is reached. Interestingly, Fan et al. found analogous results for neuronal cells cultured on SiO_2_ layers and, moreover, reported that the cells migrated to areas of optimal roughness on patterned surfaces [54]. Recently published work from Yoon et al. showed the superiority of nanostructured surfaces (carbon nanotubes) in comparison to smoother graphene substrates and polystyrene films in terms of neuronal marker expression and neural activity in multielectrode arrays, recording experiments for differentiated SH-SY5Y cells [55]. A lower differentiation-induced apoptotic rate and a higher cell proliferation rate are reported for the nanostructured materials, in comparison to the smoother surfaces. Researchers here concluded that the overall improved performance of the neuronal cells on carbon nanotube surfaces does not originate from the choice of the material (carbon), but rather from the nanoscale topography of these substrates. This article included bright-field images, which show the formation of more and larger cell agglomerations on the nanostructured material in comparison to the graphene and polystyrene films. Thus, we expect that the formation of cell clusters might be beneficial for the performance of neuronal cells—in agreement with Shipley et al., who reported that SH-SY5Y tend to grow in clusters under physiologic conditions [56]. In line with Yoon et al., we saw an improvement in cell proliferation and the formation of especially large clusters of our SH-SY5Y cells on the nanostructured TiN material in comparison to the widely used gold and ITO materials. In contrast to Yoon et al., we expect that the chemical composition of the surface strongly influences cell behavior, and it is not only the topography that determines proliferation and cell adhesion. Even though our Au and TiN substrates exhibited a very similar RMS roughness ((2.95 ± 1.63) vs. (2.98 ± 1.24) nm, respectively) and similar grain sizes ((82 ± 10) vs. (90 ± 11) nm, respectively), neurons proliferated and organized very differently on the surfaces. Thus, it is not only the structure but also the surface chemistry that determines cell behavior.

In a study conducted by Piret et al. with neuronal cells cultured on gallium phosphide materials, a cluster formation of cells was found for both flat and nanostructured (nanowire) surfaces [57]. However, for neurons cultured on flat and nanostructured silicon substrates [58], the cells formed clusters only on the flat surfaces, whereas a homogeneous cell distribution was found on the nanowires. The even cell pattern was apparently accompanied by the loss of functional neuronal network abilities. Thus, it can be concluded that the cluster formation of neurons reflects a favorable cell behavior in in vitro conditions in line with good proliferation, as also seen in our experiments. Here, nanocolumnar TiN offers the best culture conditions for SH-SY5Y cells compared to ITO and gold surfaces, and TiN with larger grain sizes. To this end, our study presents a quantitative tool to assess the neural cell organization on various surfaces in vitro by employing radial autocorrelation functions in combination with cluster analysis. Studies of cell organization can show to what extent the biomaterial supports physiologic growth conditions for specific cell types. The study of biocompatibility and bioactivity of materials in vitro can be complemented by the investigation of characteristics, such as cell morphology, cell–cell interactions, and physiological and differentiation status.

## 4. Materials and Methods

### 4.1. Electrode Materials Preparation

Cover glasses with a diameter of 13 mm and a thickness of 0.13–0.16 mm (VWR GmbH, Darmstadt, Germany) with four different coatings were employed to deposit thin films of indium tin oxide (ITO), gold (Au), and titanium nitride (TiN) in two different surface topographies.

Before film deposition, all coverslips were cleaned in a standard procedure with acetone and isopropanol in an ultrasonic bath and subsequently in 3% hydrofluoric acid for 2 min, and then rinsed with ultra-pure water. The metallic coatings were applied by using the sputter process (CREAMET 500, Creavac GmbH, Dresden, Germany). For the ITO plating, a 4” indium tin oxide target (90:20 wt.%, EVOCHEM GmbH, Offenbach am Main, Germany) was used at a working pressure of 4.5 × 10^−3^ mbar with an argon (Ar) flow rate of 18 sccm, a combined power of 250 W (DC) and 85 W (RF), and a working distance of 150 mm for 20 min. A heat treatment at 400 °C was performed for 10 min to increase the transparency of the layers. For the gold plating, it was necessary to apply an adhesion promoter layer before, which was realized by the deposition of a 50 nm layer of indium tin oxide. The subsequently applied gold layer was produced at 4.5 × 10^−3^ mbar, 350 W (DC), a working distance of 150 mm, and an argon flow rate of 18 sccm with a 4” gold target (99.99%, Heimerle&Meule GmbH, Pforzheim, Germany) for 3 min. The same ITO target and sputtering parameters in terms of working distance, working pressure, and argon flow rate were also employed to gain the ITO surfaces for later cell experiments. Here, the sputtering time was 3 min at 350 W (DC).

In order to produce two different titanium nitride (TiN) layers with different topographies, first, titanium (Ti) was used as an adhesion promoter with a 4” titanium target (99.99%, Kurt J. Lesker Company, Jefferson Hills, PA, USA) at 4.5 × 10^−3^ mbar with an Ar flow rate of 18 sccm, power of 500 W (DC), working distance of 150 mm, and a sputtering time of 5 min. Afterward, a gold layer was coated on top, as described above. Subsequently, the titanium nitride layer was produced by sputter deposition with the same titanium target used before at the same pressure and working distance. In addition to the process, nitrogen with a purity of 99.95% was added at a flow rate of 6 sccm, whereby the Ar flow rate was reduced to 11 sccm. The sputtering power during the process was 600 W (DC). For the thin TiN layers, a coating time of 2.5 min was chosen, and 40 min for the thick layers. The thick layer is defined as TiN nano or nanocolumnar TiN in the Results section.

### 4.2. Atomic Force Microscopic Analysis

Surface morphologies of the electrode materials were imaged using a JPK NanoWizard 3 atomic force microscope (Bruker Nano GmbH, Berlin, Germany). Data acquisition was performed in direct drive AC mode with a TESPAHAR cantilever (fnom = 320 kHz, dnom = 42 N/m, Bruker). The height (measured) channel was used for image analysis in Gwyddion 2.55. Image analysis included leveling data by mean plane subtraction and, subsequently, a row alignment using the program in-built median of differences method. The root-mean-square roughness and surface areas of the samples were calculated by the statistical quantities tool. For determination of grain sizes, diameters of defined grains were manually measured using the point-to-point distance tool, measuring at least 30 grains/image.

Statistical analyses were performed using GraphPad Prism 5.02. Multiple group comparisons were performed by 1D ANOVA with Tukey’s post-hoc test. Differences between two means with *p* < 0.05 were considered significant (*), *p* < 0.01 very significant (**), and *p* < 0.001 extremely significant (***).

### 4.3. Cell Lines and Cell Culture

We chose two human brain cell lines for our studies: The human neuroblastoma cell line SH-SY5Y (Cat.No. CRL-2266, ATCC LGC Standards GmbH, Wesel, Germany) and the human primary glioblastoma cell line U-87 MG (Cat.No. 300367, CLS Cell Lines Service GmbH, Eppelheim, Germany).

U-87 MG and SH-SY5Y cells were cultured in 1:1 MEM Eagle/Ham’s F12 medium containing Earle’s salts, L-glutamine, and sodium bicarbonate (Cat.No. M4655 and N6658, Sigma-Aldrich Chemie GmbH, Munich, Germany) supplemented with 10% fetal bovine serum (Cat.No. S0615, Biochrom GmbH, Berlin, Germany) and 1% penicillin/streptomycin (Cat.No. P0781, Sigma-Aldrich Chemie GmbH, Munich, Germany). Both cell lines were incubated in separate culture flasks at 37 °C in a 95% air and 5% CO_2_ atmosphere. Cell culture medium was changed every 2–3 days. A mixture of phosphate-buffered saline (PBS, Cat.No. 18912014, Gibco Thermo Fisher Scientific, Waltham, MA, USA), 0.025% (*w/v*) trypsin, and 0.011% (*w*/*v*) ethylenediaminetetraacetic acid (EDTA, Cat.No. L2143, Biochrom GmbH, Berlin, Germany) was applied for 3–4 min to detach the cells prior to cell counting and seeding.

### 4.4. Cell Staining and Imaging

Cells were counted in an automatic optical cell counter prior to cell seeding onto the substrate materials (EVETM, NanoEntek Inc., Seoul, Korea). Subsequently, cells were seeded onto different substrate materials (Au, ITO, TiN, nanocolumnar TiN) at a density of 130 cells/mm^2^ in cell culture medium. U-87 MG cells were fixed 24 or 72 h after seeding, respectively. We did not employ longer culture times, to avoid the formation of very dense cell layers for which a cluster analysis would not be possible. Cells were fixed with paraformaldehyde (Cat.No. HT5011, Sigma-Aldrich Chemie GmbH, Munich, Germany) for 15 min. In order to fluorescently label actin fibers and cell nuclei, cells were washed with PBS and cell membranes were permeabilized with a PBS solution containing 1% (*w/v*) Triton X-100 (Cat.No. 9002-93-1, Sigma-Aldrich Chemie GmbH, Munich, Germany) and 0.5% (*w/v*) bovine serum albumin (Cat.No. A2153, Sigma-Aldrich Chemie GmbH, Munich, Germany) for 10 min at room temperature. Afterward, cells were incubated with a PBS solution supplemented with 1 µg/mL Hoechst 34,580 (Cat.No. H21486, Molecular Probes, Eugene, OR, USA) and 0.44 µM Alexa Fluor 532 Phalloidin (Cat.No. A-22282, Molecular Probes) at room temperature for 15 min. Subsequently, substrates with fixed cells were washed again with PBS and placed upside down into clean Petri dishes (Cat.No. 80136, ibidi GmbH, Gräfeling, Germany) with mounting medium (Cat.No. 50001, ibidi GmbH, Gräfeling, Germany) to prepare the samples for imaging in an inverse confocal laser scanning microscope. Samples were stored at 4 °C until imaging.

For the SH-SY5Y cells, 24 h after seeding onto the different substrate materials, cells were supplemented with 25 nM staurosporine (Cat.No. S5921, Sigma-Aldrich Chemie GmbH, Munich, Germany) to initiate the cell differentiation process, which takes 72 h to complete [59]. The first samples of SH-SY5Y cells for each substrate type were fixed directly upon removing the culture medium containing the staurosporine, while the second samples were cultured for another 72 h in growth medium and fixed afterward as described above. After fixations, actin fibers and cell nuclei were fluorescently labeled according to the U-87 MG cells.

Cell network morphology was investigated using confocal laser scanning microscopy. Images were acquired with an inverted Zeiss Axio Observer.Z1 microscope equipped with a spinning disk unit (Yokogawa CSU-X1A 5000, Tokyo, Japan) and a 25 × glycerin immersion objective. The complete cell network for each substrate material was imaged as an array of individual dual-channel fluorescence images. Each image encompassed a substrate area of 0.22 cm^2^. Thus, up to 54 images were required to image the entire substrate area.

### 4.5. Image Analysis and Autocorrelation of Cell Positions

Images of neuronal cell networks were processed using Fiji distribution [60] (Windows 10, 64-bit version) based on ImageJ software [61]. The position of the cells was identified by the location of fluorescently labeled cell nuclei using the particle tracker tool on previously thresholded and binarized images. Thus, for all images, we extracted the following parameters for each cell type and substrate: Number of cells, growth area, and mean cell density.

In order to analyze the cellular network organization in terms of the nearest neighbors of each cell, the macro “Radially Averaged Autocorrelation” combined with the “Radial Profile” plugin was employed to evaluate a radially averaged two-point autocorrelation function S2 for all images. Such analysis allows measurement of the average size of objects (patches of cell clusters) in conjunction with the distance between these objects as similarly shown by Baker et al. [46] and described in detail by Berryman et al. [62]. Briefly, the ImageJ plugin computes the probability of finding a black pixel in increasing radial distance to an initially chosen black pixel. This process is repeated multiple times with different initial pixels. The results are radially averaged in a second step. The chosen plugin utilizes a fast Fourier transform (FFT) to reduce computation time, while simultaneously correcting for the periodicity of the FFT and finite image size, so the results do not suffer from artifacts. The results are normalized such that the value of the radially averaged autocorrelation function will always be 1 (perfect correlation) at a distance *r* = 0. It directly follows that an output value of 0 demonstrates the case of no correlation.

### 4.6. Cluster Analysis of Cellular Network Organization

Additionally, we employed self-written cluster analysis tools programmed in R for further investigation of the spatial distribution of cells on the substrates, including network patterns such as cluster formation. The goal of these cluster analysis methods is to minimize the within-cluster variation. In other words, the clusters are supposed to be dense but located far apart. The within-cluster variation *W* is defined as the sum of squared distances between cells and their corresponding centroid (i.e., cluster center):(1)$$W\left(C_{k}\right)=\underset{x_{i}\in C_{k}}{\sum }∥x_{i}-\mu_{k}{∥}^{2}$$
where $x_{i}$ refers to a cell position in the corresponding cluster $C_{k}$ with $\mu_{k}$ being the mean value of all data points assigned to this cluster. The total within-cluster variation for any given data set (cell image) is then defined as:(2)$$totalwithin=totW\left(k\right)=\overset{K}{\underset{k=1}{\sum }}W\left(C_{k}\right)$$
This function needs to be minimized in order to make the clusters as compact as possible.

Our analysis is based on a *K*-means clustering algorithm [63]. It groups the positions of cell nuclei iteratively from a previously set number of clusters *K*. Initially, the positions of the cluster centers are chosen randomly. The distance of each cell to its nearest cluster center is calculated using the Euclidean norm. The algorithm then calculates iteratively new positions of cluster centers to optimize the distances of all cells to their assigned cluster centers, while the number of clusters *K* is kept constant. Thus, the position of the cluster centers changes in every iteration of the algorithm. The sorting of cells into clusters is finished once the cluster center positions stabilize. The outcome of the *K*-means algorithm and the quality of the results depend highly on the initial choice of the number of clusters *K*. Therefore, we performed gap statistics and used the elbow method to verify the credibility of our choice of *K*.

Determination of the number of cell clusters (1): The optimal number of clusters for each specimen is obtained by employing the elbow method [64]. The *K*-means algorithm is run for several different numbers of clusters, i.e., values of *k*. The total within-cluster sum of squares totW(k) is calculated as indicated in the equations above from the sum of squared distances of data points in the cluster and its centroid for all *k* values. We then find the appropriate *k* value, i.e., the optimal number of clusters, by plotting totW(k) as a function of *k* (see Appendix A, Appendix A). This curve starts to flatten at some point and forms an “elbow”, which is regarded as an indicator of the optimal number of clusters. To avoid possible deviations by the correct choice of *K* from the resulting curves, we compared our results to gap statistics to check our choice of *K*.

Determination of the number of cell clusters (2): To confirm our results drawn from the elbow method, we additionally used a gap statistic tool in our analysis code. The gap statistic is based on the comparison of within-class variation of real data and what we would expect from a hypothetical uniform data set [65]. To this end, we compared the total within-cluster variation totW(k) of our cell network data with a model of uniformly distributed data points. The Gap for *k* clusters is calculated as follows:(3)$$Gap\left(k\right)=logtotW_{uni}\left(k\right)-logtotW\left(k\right)$$
The total within-cluster variation $totW_{uni}\left(k\right)$ of the uniform data set is found by a computer simulation of 50 uniformly distributed data sets. The code also measures the standard error s(k) associated with the simulated data sets, and gives the optimal number of clusters *K* based on this equation:(4)$$K=min\{k\in \{1,\ldots ,k_{max}\}:Gap\left(k\right)\geq Gap\left(k+1\right)-s\left(k+1\right)\}$$
We follow the suggestion of Tibshirani et al. [65] and set $s\left(k\right)=\sqrt{1+\frac{1}{50}}sd\left(k\right)$ where *sd(k)* denotes the standard deviation of the 50 Monte-Carlo-simulated data sets.

A graphical illustration of the value Gap(k) as a function of different *k* values is shown in Appendix A, Appendix A. The optimal number of clusters is represented by the smallest value of *k*, where the above-mentioned inequality is fulfilled. That means we maximize the Gap value such that Gap(k) is within one standard deviation of the Gap at *k* + 1.

Cellular grouping within clusters: In addition to these two techniques, we used the silhouette method to check and verify the quality of the grouping of cells into certain clusters [66]. It delivers one of three outcomes of the silhouette coefficient SilCoef for each cell: −1, 0, or +1. The value 0 means the cell is positioned at the edge of the cluster, −1 corresponds to the cell being assigned to the wrong cluster, and +1 means we sorted the cell into the correct cluster. The silhouette coefficient is calculated for every cell *i* as follows:(5)$$SilCoef\left(i\right)=\frac{x\left(i\right)-y\left(i\right)}{max\{x\left(i\right),y\left(i\right)\}}$$
where x(i) depicts the smallest mean distance to cells in any other cluster, and y(i) is the mean intracluster distance. A high average silhouette width indicates a good clustering result. Thus, the silhouette method can also be used to check the chosen optimal number of clusters in our experiments. We plotted the mean silhouette width as a function of various values of clusters *k*. The location of the maximum of that curve is considered to be the optimal number of clusters for the experiment.

From the cluster analysis, the following parameters were extracted for both cell types and all employed substrate materials: The number of cells in each cluster, cluster area, and spatial cell density within clusters.

### 4.7. Statistical Analysis of Cell Analysis

For both cell types, three specimens per substrate were made for all combinations of the four substrate materials and both cell growth times. In total, we analyzed around 160,000 cells for our investigations.

Averages of data within individual samples or independent experiments were expressed as the arithmetic mean ± standard error of the mean. Statistical significance between data sets was evaluated using the two-sample *t*-test tool in OriginLab software (OriginPro 2017G, OriginLab Corporation). Values differing by *p* ≤ 0.05 were significant, and values differing by *p* ≤ 0.01 were considered as highly significant.

## 5. Conclusions

The importance of in vitro neuron and glial cell experiments as preliminary tests for not only in vivo applications such as neuroelectrodes [67,68,69] but also as stand-alone research, e.g., to investigate neural activity and neurodegenerative diseases on lab-on-a-chip devices such as multielectrode arrays (MEA), has become apparent lately [70]. This technique has been continuously improved over the past decade [38] and offers great potential to shift expensive and ethically questionable in vivo animal experiments to cost-efficient and easy-to-use high-throughput in vitro assays.

In our study, we showed that fluorescent imaging, to identify the size and position distribution of cell agglomerations in combination with a proliferation assay, is a quantitative tool to measure the biocompatibility of novel biomaterials. In fact, the combination of the K-means clustering algorithm and calculation of the radially averaged autocorrelation function is able to identify the whole range of cell patterns from large dense clusters down to individual and homogeneously distributed cells, giving an individual fingerprint for different cell types. Here, we found nanocolumnar TiN surfaces to perform best in terms of cell division and the network formation characteristic for SH-SY5Y cells. Future studies will focus on the question of how cellular organization on nanocolumnar TiN surfaces correlates with the physiological status of neurons and glial cells. As thin films of nanocolumnar TiN exhibit excellent bioactive properties in combination with optical transparency and low electric resistance, the application of this material in multielectrode arrays will be tested soon.

## Figures and Tables

**Figure 1 ijms-21-06249-f001:**
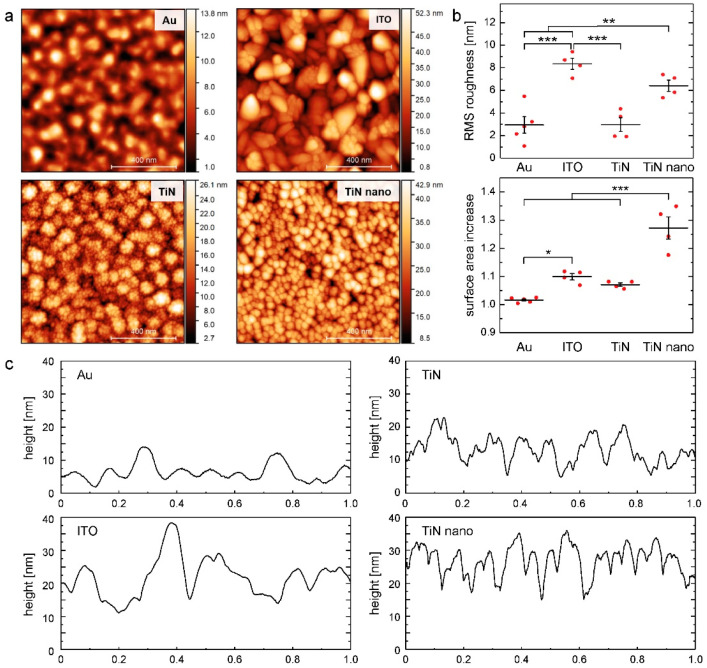
Atomic force microscopy characterization of the tested electrode materials: (**a**) 1 × 1 µm images of gold (Au), indium tin oxide (ITO), titanium nitride (TiN), and TiN nano with nanocolumnar structure; (**b**) area-derived metrics of the AFM images (root-mean-square (RMS) roughness and ratio of surface area to projected area (surface area increase)); the horizontal lines represent the mean values of the data points (red dots), while the vertical lines show the standard errors (mean ± se, ANOVA with Tukey’s post-hoc test, * = *p* < 0.05, ** = *p* < 0.01, *** = *p* < 0.001); (**c**) one-line profiles of AFM images.

**Figure 2 ijms-21-06249-f002:**
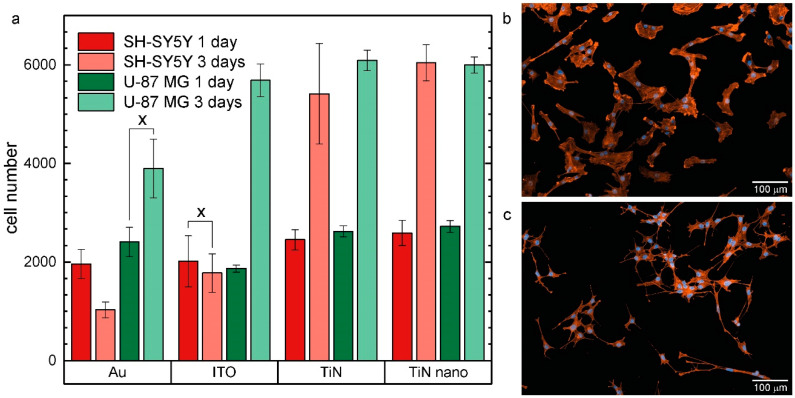
(**a**) Average number of SH-SY5Y and U-87 MG cells grown on different electrode materials (Au, ITO, TiN, nanocolumnar TiN) after one and three days in culture. Values marked with x are not statistically significant (*p* > 0.05); (**b**) fluorescent image of U-87 MG cells cultured on TiN nanocolumnar surfaces for 1 day. Cell nuclei are blue and actin fibers are colored orange. The scale bar represents a length of 100 µm; (**c**) fluorescent image of SH-SY5Y cells grown on a TiN nanocolumnar substrate for 1 day plus additional 72 h incubation with culture medium supplemented with staurosporine to induce cell differentiation. Colors and scale bar as in (**b**).

**Figure 3 ijms-21-06249-f003:**
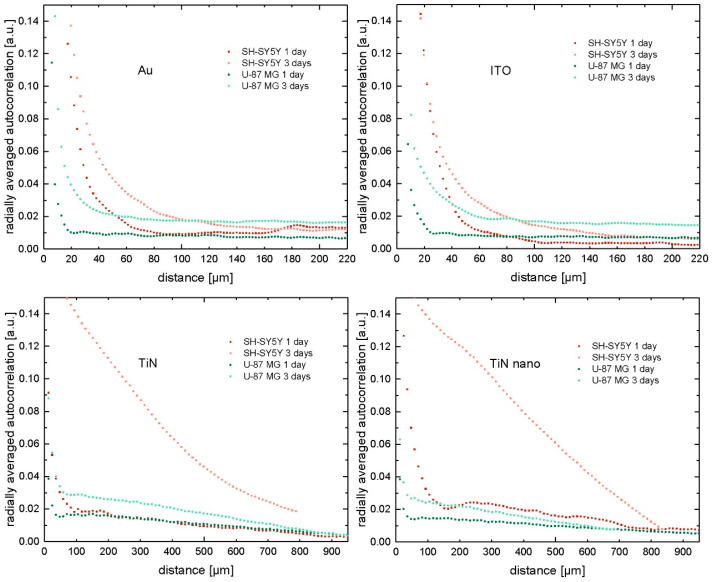
Radially averaged autocorrelation analysis for cell nuclei positions obtained from 12 individual experiments with U-87 MG cells (green) and SH-SY5Y cells (red) cultured on different materials.

**Figure 4 ijms-21-06249-f004:**
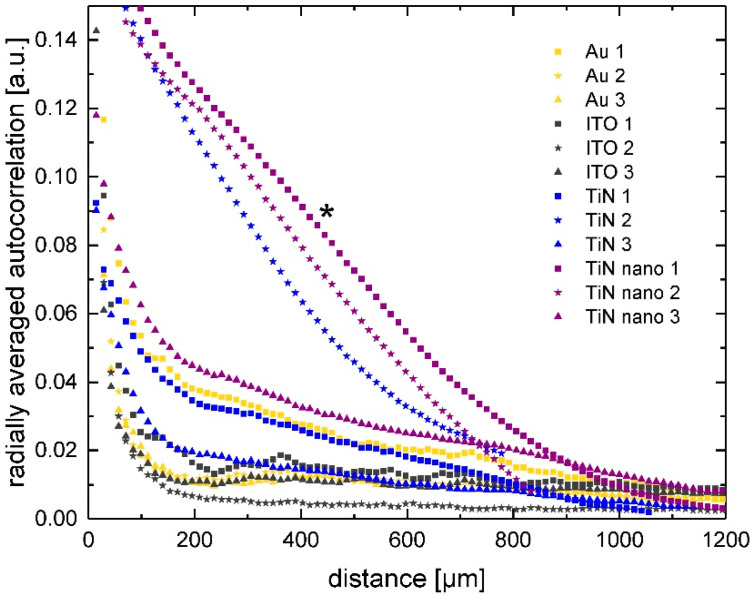
Radially averaged autocorrelation function of cell nuclei positions of SH-SY5Y neuronal cells after 3 days of culture on various substrate materials. The experiment and data points marked with * serves as an example of the K-means cluster analysis shown in Figure 5 and Figure 6.

**Figure 5 ijms-21-06249-f005:**
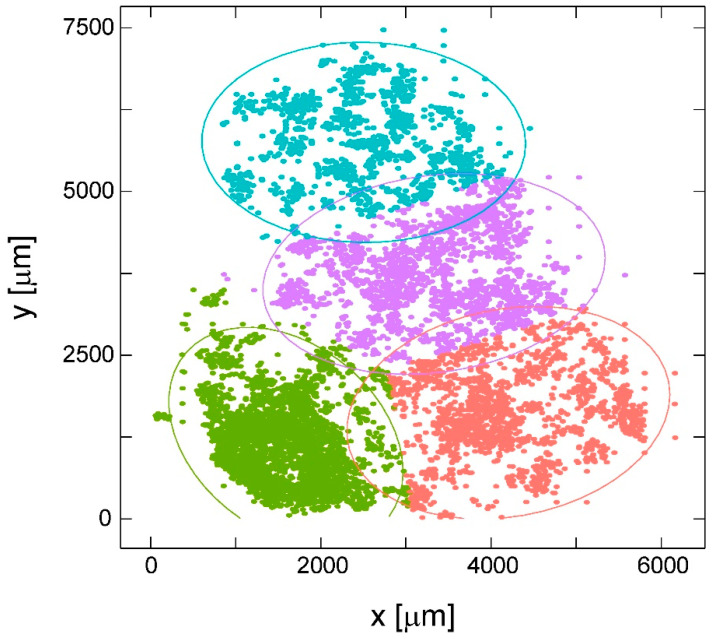
Result of K-means clustering of SH-SY5Y cells grown on TiN nanocolumnar substrates after 3 days. Each data point represents a single cell. The different colors denote cluster 1 (red), cluster 2 (green), cluster 3 (blue), and cluster 4 (purple). The cell distribution corresponds to the autocorrelation function marked with * in Figure 4. Note that the graph does not represent the entire substrate area of 0.22 cm^2^, but only the area covered by cells.

**Figure 6 ijms-21-06249-f006:**
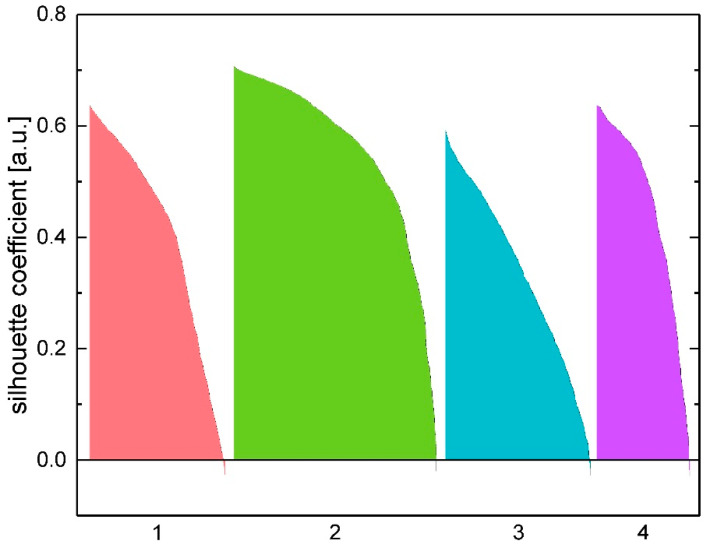
Silhouette width plot associated with the example experiment shown in Figure 5 (neuronal cells on TiN nanocolumnar surfaces after 3 days). The four clusters are color-coded. Positive values indicate a high-quality cell sorting result.

**Figure 7 ijms-21-06249-f007:**
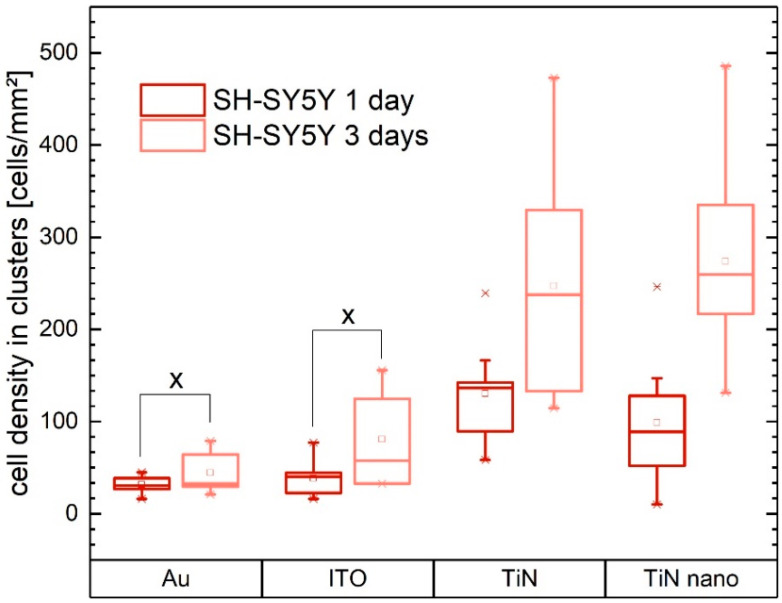
Cell density in clusters of SH-SY5Y cells grown on different electrode materials (Au, ITO, TiN, TiN nano) after one and three days of growth as a result of K-means clustering. Values marked with X are not statistically significant (*p* > 0.05).

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
