# Peer review of "Proliferation and Cluster Analysis of Neurons and Glial Cell Organization on Nanocolumnar TiN Substrates"

_ijms, 2020, doi:10.3390/ijms21176249_

Round 1

Reviewer 1 Report

In this study the authors compared the growing behaviors of neurons and glial cells on different substrates, including Au, ITO, thin TiN layer and thick TiN layer (TiN nano). However, the rationale of comparing the growing behavior of neural and glial cells on different inorganic materials, such as Au, ITO, TiN and TiN nano-columnar is not clear in this study.

In this study the authors also proposed to quantify the organization and distribution of cells on different substrates by using radial autocorrelation function of cellular position in combination with different clustering algorithms to try to find out whether these cells are homogeneously spread or agglomerate on the surface of the specific materials. Although cell viability and organization are two important features, they are not enough to verify the growing status of the cultured cells and the biocompatibility of the materials. Cell morphology, cell physiological properties are also important for the determination of the viability and physiological status of cells. This is especially true for neural cells, which exhibit different biological and physiological properties, morphologies and connectivity under different conditions and different differentiation states. Hence, it is not enough to evaluate and justify the biocompatibility of biomaterials by simply via the proliferation and spatial organization of the cultured cells.

In fact, the organization and distribution of cells on the surface of either organic or inorganic materials could be affected by many factors, including cell-substrate interactions, cell-cell interactions, bioactivity of the materials (neutral, positive or negative), surface roughness of the materials and migratory capability of cells. SH-SY5Y cell line is special due to its differential growing behaviors before and after differentiation. The SH-SY5Y cells tend to grow in clusters at their undifferentiated state. After differentiation, SH-SY5Y cells start to migrate and spread from the clusters. Hence, it is not surprised to find that the SH-SY5Y cells tend to grow in cluster on all four materials (Figure 3), when they were induced to differentiate into neural form after seeding on the substrates.

Honestly, the main question of this study is not how SH-SY5Y and U-87 MG cells grow and organize on the surface of the materials, but how the surface roughness and topology of materials affect the growth of SH-SY5Y and U-87 MG cells. Without the support of the physiological status of cells the information about the organization and distribution of cells is meaningless. In reviewer’s opinion Au exhibited a severe cytotoxicity to both SH-SY5Y and U-87 MG cell lines. ITO exhibits severe cytotoxic effect to SH-SY5Y cells but a minor effect to U-87 MG cells. Unfortunately, the authors did not follow this lead to solve the problem. In addition, a strong interaction may occur between TiN and SH-SY5Y cells that leads to the appearance of large cell agglomerations in SH-SY5Y cell population. The authors should discuss these issues in the manuscript.

Specific comments:

1) There are many grammatical errors in this manuscript. A thorough English proofread and editing is required.

2) In the Introduction section the authors talked a lot about the materials adopted for the development of electrodes for electrochemical sensors and biosensor, but give little or no literature review about the materials that were used in the development of devices for cell culturing. More literature review about the MEMS devices, e.g., lab-on-a-chip, should be included in the Introduction section.

3) The rationale of studying and comparing the cell growing behavior on different inorganic materials, such as Au, TIO, TiN and TiN nano-columnar should be described and justified in the manuscript.

4) How long were SH-SY5Y cells grown on the TiN nanocolumnar in Figure 2c?

Author Response

Reviewer 1

In this study the authors compared the growing behaviors of neurons and glial cells on different substrates, including Au, ITO, thin TiN layer and thick TiN layer (TiN nano). However, the rationale of comparing the growing behavior of neural and glial cells on different inorganic materials, such as Au, ITO, TiN and TiN nano-columnar is not clear in this study.

We thank the reviewer for taking the time to carefully read our manuscript and sending this detailed reply. Especially the fast response is highly appreciated.

We carefully addressed all issues. All changes in the manuscript are marked in yellow.

Referring to the rationale of our study, please see our comments below (specific comments no 3 where the reviewer asked the same question).

In this study the authors also proposed to quantify the organization and distribution of cells on different substrates by using radial autocorrelation function of cellular position in combination with different clustering algorithms to try to find out whether these cells are homogeneously spread or agglomerate on the surface of the specific materials. Although cell viability and organization are two important features, they are not enough to verify the growing status of the cultured cells and the biocompatibility of the materials. Cell morphology, cell physiological properties are also important for the determination of the viability and physiological status of cells. This is especially true for neural cells, which exhibit different biological and physiological properties, morphologies and connectivity under different conditions and different differentiation states. Hence, it is not enough to evaluate and justify the biocompatibility of biomaterials by simply via the proliferation and spatial organization of the cultured cells.

We thank the reviewer for the valuable comment that investigating “only” cell proliferation and cellular organization on a biomaterial is not enough to evaluate biocompatibility. In our study, we aim to show how cellular organization can be quantified employing computational tools in combination with proliferation assays as one step to assess biocompatibility in vitro.

Studies of cell organization can show to what extent the biomaterial supports physiologic cell organization on its surface as a first step to develop new materials for improved neuron-electrode interaction. Especially for the employed SH-SY5Y cell line, it is known that cluster formation occurs when these cells are cultured under physiologic conditions as discussed in the manuscript. Thus, this study should offer new tools to mathematically quantify cellular behavior on the example of different materials.

To make our statements clearer, we added additional information to our manuscript on page 2, lines 94-95 and pages 11-12, lines 373-379.

In fact, the organization and distribution of cells on the surface of either organic or inorganic materials could be affected by many factors, including cell-substrate interactions, cell-cell interactions, bioactivity of the materials (neutral, positive or negative), surface roughness of the materials and migratory capability of cells. SH-SY5Y cell line is special due to its differential growing behaviors before and after differentiation. The SH-SY5Y cells tend to grow in clusters at their undifferentiated state. After differentiation, SH-SY5Y cells start to migrate and spread from the clusters. Hence, it is not surprised to find that the SH-SY5Y cells tend to grow in cluster on all four materials (Figure 3), when they were induced to differentiate into neural form after seeding on the substrates.

We fully agree that the organization of cells depends on the type of material, surface properties, chemistry, differentiation, etc. In our study, we focus on the question of how this organization can be quantified and introduce computational tools to compare cell behavior on different surfaces. In fact, we see that SH-SY5Y cells tend to grow in clusters on all four materials (Figure 3) as also stated by the reviewer; however, the sizes of the clusters strongly vary, as well as the overall distributions of these cells on the different surfaces. Here we show how the different cluster sizes and cellular distributions change depending on the surface they are growing on. These differences together with the proliferation data are then used to propose which materials exhibit the best biocompatibility features.

Honestly, the main question of this study is not how SH-SY5Y and U-87 MG cells grow and organize on the surface of the materials, but how the surface roughness and topology of materials affect the growth of SH-SY5Y and U-87 MG cells. Without the support of the physiological status of cells the information about the organization and distribution of cells is meaningless.

We agree that the main question of our manuscript is twofold: First, we aim to address how structural organization of cells on surfaces can be quantified and, second, we show how the employed algorithms can be used to quantify cell behavior on different surfaces. Here the interaction of neurons and glial cells with different materials acts as a proof of principle of our analysis tool.

Further evaluation of how cell organization correlates with the physiological status of the cells is beyond the scope of our study. Our current investigations (when the labs will be fully accessible again due to the Corona crisis) focuses on exactly this question, together with the detailed study on cell adhesion forces (determined by single-cell force spectroscopy) for different culture times.

We added a respective statement in the Conclusion on page 16, lines 568-569.

In reviewer’s opinion Au exhibited a severe cytotoxicity to both SH-SY5Y and U-87 MG cell lines. ITO exhibits severe cytotoxic effect to SH-SY5Y cells but a minor effect to U-87 MG cells. Unfortunately, the authors did not follow this lead to solve the problem. In addition, a strong interaction may occur between TiN and SH-SY5Y cells that leads to the appearance of large cell agglomerations in SH-SY5Y cell population. The authors should discuss these issues in the manuscript.

We employed gold and ITO substrates as controls for our investigations since these materials are already employed for in vitro and in vivo applications in contact with neurons and glial cells for decades. Both materials are considered to be non-toxic in contact with neurons and glial cells (see reference: DOI 10.3390/ma11101995). However, we fully agree that there are drawbacks to these materials in terms of possible cytotoxicity. That is why we addressed nanocolumnar TiN and its interaction with neurons and glial cells as a possible alternative for future applications. In terms of TiN, large cell agglomerations in combination with a positive proliferation behavior is employed here as a tool to evaluate biocompatibility since agglomeration of these cells is seen under physiological conditions, while a different cellular organization is expected to occur e.g. when cytotoxic effects come into play.

We added a statement in the introduction of why gold and TiN were used in our study (page 3, lines 97-98 and 100-102).

Additionally, we agree that strong interactions may occur between TiN and SH-SY5Y cells that lead to the appearance of large cell agglomerations in SH-SY5Y cell population. To address this question in more detail and to check how cell-surface interaction correlates with cellular organization on the different surfaces, we are currently investigating cell adhesion on these materials as a function of culture time and surface properties with different methods including single-cell force spectroscopy. Here we also employ co-culture methods to address the question mentioned by the reviewer: why do some materials exhibit stronger cytotoxic effects on SH-SY5Y than on U-87 MG cells. However, this study is beyond the aim of our manuscript.

Specific comments:

1) There are many grammatical errors in this manuscript. A thorough English proofread and editing is required.

Our author Chelsie Steele is a native speaker and checked our manuscript carefully before the first submission. Now, we additionally used a professional computer program for grammar checks and corrected the sentences on the following pages and lines:

Page 1 lines 23-24

Page 1 lines 26-29

Page 3 lines 103-106

We also corrected commas.

 2) In the Introduction section the authors talked a lot about the materials adopted for the development of electrodes for electrochemical sensors and biosensor, but give little or no literature review about the materials that were used in the development of devices for cell culturing. More literature review about the MEMS devices, e.g., lab-on-a-chip, should be included in the Introduction section.

As suggested by the reviewer, we added a statement on MEMS, as well as neural circuits on a chip. Lab-on-a-chip devices such as multi-electrode arrays were already included in the introduction.

The new statements are included on page 2, lines 74-78.

3) The rationale of studying and comparing the cell growing behavior on different inorganic materials, such as Au, TIO, TiN and TiN nano-columnar should be described and justified in the manuscript.

In our study, we investigated the interaction of neurons and glial cells with nanocolumnar TiN, since this material exhibits an increased surface area, ideal for miniaturized multielectrode arrays. In fact, the increased surface area allows the shrinking of microelectrode size without losing detection sensitivity due to a lowered self-impedance of the electrode. Gold and ITO substrates were used for control experiments since these materials are often the materials of choice for electrodes in contact with neurons (see reference: DOI 10.3390/ma11101995).

To make our statements clearer, we added the respective information in the introduction on page 3, lines 97-99 and 101-103.

4) How long were SH-SY5Y cells grown on the TiN nanocolumnar in Figure 2c?

We thank the reviewer for this valuable comment. We added the respective information to the caption of Figure 2 for clarification (lines 165-166):

“Fluorescent image of SH-SY5Y cells grown on a TiN nanocolumnar substrate for 1 day plus additional 72 h incubation with culture medium supplemented with staurosporine to induce cell differentiation.”

Reviewer 2 Report

In this work authors describe the analysis of neuronal and glial cell growth on TiN substrates and compare it to cell growth on gold and ITO substrates with larger substrate surface microstructures. Although for the first day the difference is minor, for the third day a substantial enhancement of growth for TiN substrates was observed. It is critical that authors also developed a statistical methods of assessing cell cluster formation as neuronal cell clusters were shown to be critical for their functionality. As determined by the authors, TiN cells yield larger agglomerates suggesting a high benefit of such substrates for applications in biosensors and bio-microelectronics. This finding as well as the  statistical cluster assessment method developed by the authors is highly important and should be published with just a few minor questions to be addressed:

  • I suggest authors to mention in the introduction of any other TiN substrate cell growth studies that were performed, otherwise outline uniqueness of their work.
  • Second, I suggest to comment on general cell growth on all the substrates farther beyond 3 days and if the trend observed persists (data is not required, just a general observation would be beneficial to the reader). 

Author Response

Reviewer 2

In this work authors describe the analysis of neuronal and glial cell growth on TiN substrates and compare it to cell growth on gold and ITO substrates with larger substrate surface microstructures. Although for the first day the difference is minor, for the third day a substantial enhancement of growth for TiN substrates was observed. It is critical that authors also developed a statistical methods of assessing cell cluster formation as neuronal cell clusters were shown to be critical for their functionality. As determined by the authors, TiN cells yield larger agglomerates suggesting a high benefit of such substrates for applications in biosensors and bio-microelectronics. This finding as well as the statistical cluster assessment method developed by the authors is highly important and should be published with just a few minor questions to be addressed:

First of all, we would like to thank the reviewer for the time to carefully read our manuscript and the positive report which is highly appreciated. All changes in the manuscript are marked in yellow.

  • I suggest authors to mention in the introduction of any other TiN substrate cell growth studies that were performed, otherwise outline uniqueness of their work.

We fully agree with the reviewer and thank him/her for the valuable comment. The employed nanocolumnar TiN substrates are new and have not been investigated in contact with cells yet. Thus, we added a respective statement into the introduction (page 3, lines 97-99)

  • Second, I suggest to comment on general cell growth on all the substrates farther beyond 3 days and if the trend observed persists (data is not required, just a general observation would be beneficial to the reader). 

The reviewer raises the very interesting question of long-term culturing effects here. This should definitely be addressed in future experiments.

One important reason why we did not employ longer culture times was the possibility to compare different surface patterning in terms of how cells spread and organize on the different surfaces. Longer culture times, however, resulted in very dense cell layers due to cell division and cluster analysis was not possible anymore since cells just grew into free space until the entire surface was covered. Anyway, we fully agree that longer culture times are important for a better understanding of cell-surface interaction. Thus, we have already started a new research project in which we correlate cell organization on the same substrates with adhesion forces (determined by single-cell force spectroscopy) as a function of culture time to gain more insides into this interesting topic.

We added a respective statement on page 13, lines 439-441.

Round 2

Reviewer 1 Report

The main purpose of this study is to quantify the growth behavior and spatial organization of cells on the surface of materials to quantify and evaluate the performance of cells on the potential materials. The authors developed a method by combining radial autocorrelation function and various clustering algorism to try to quantify the organization and distribution of cells on the surface of various materials, e.g., Au, ITO, TiN and TiN nanocolumnar. The authors proposed that, with the information about the cell proliferation and the cell distribution and organization, one can evaluate the performance of the potential (bio)materials.

It is fine for the authors to propose to determine the cell spatial organization and clustering by the method and algorism established in this study. However, the main purpose of doing this should be carefully defined. It is for sure that the biocompatibility and the performance of a material or a device cannot be simply by observing the cell growth and the spatial organization of cells on their surface. This is especially true for those novel or unstudied materials or devices, i.e., TiN nanocolumnar. Cells may grow and spread on the surface of a material normally. However, inside the cells, they may suffer from the oxidative stress, autophage, transformation, differentiation, growth alternation, internal reorganization, etc.

Specific comments:

1) In p.3 (lines 101-103), the authors mentioned that “ ..these materials (e.g., gold and ITO) are considered to be non-toxic and are often the materials of choice for electrodes in contact with neurons [45].” However, in this study the results they found on gold and ITO were just opposite to what was reported previously. The authors should explain and discuss this problem in the manuscript.

2) From the studies in Section 2.2, the authors concluded that TiN and TiN nanocolumnar are suitable for the growth of neuron and glial cells. However, the authors did not compare their results with those growing neurons under the conventional methods. In this case, the authors are recommended to compare the results in the reports regarding the culturing of SH-SY5Y and U-87 MG.

Author Response

The main purpose of this study is to quantify the growth behavior and spatial organization of cells on the surface of materials to quantify and evaluate the performance of cells on the potential materials. The authors developed a method by combining radial autocorrelation function and various clustering algorism to try to quantify the organization and distribution of cells on the surface of various materials, e.g., Au, ITO, TiN and TiN nanocolumnar. The authors proposed that, with the information about the cell proliferation and the cell distribution and organization, one can evaluate the performance of the potential (bio)materials.

It is fine for the authors to propose to determine the cell spatial organization and clustering by the method and algorism established in this study. However, the main purpose of doing this should be carefully defined. It is for sure that the biocompatibility and the performance of a material or a device cannot be simply by observing the cell growth and the spatial organization of cells on their surface. This is especially true for those novel or unstudied materials or devices, i.e., TiN nanocolumnar. Cells may grow and spread on the surface of a material normally. However, inside the cells, they may suffer from the oxidative stress, autophage, transformation, differentiation, growth alternation, internal reorganization, etc.

We thank the reviewer for carefully checking our answers to her/his questions. The fast response is highly appreciated.

We addressed all remarks and highlighted all changes in the manuscript in yellow.

We agree that there is more to biocompatibility of biomaterials than the observation of cell growth on substrates. We will explore this issue in further experiments and address the physiological state of the cells as mentioned in our manuscript. Our current study aims to introduce computational tools in combination with proliferation assays as a new step to assess cell growth and spatial organization in vitro.

Considering possible cytotoxic effects, please see our next comment.

Specific comments:

1) In p.3 (lines 101-103), the authors mentioned that “ ..these materials (e.g., gold and ITO) are considered to be non-toxic and are often the materials of choice for electrodes in contact with neurons [45].” However, in this study the results they found on gold and ITO were just opposite to what was reported previously. The authors should explain and discuss this problem in the manuscript.

We thank the reviewer for this valuable comment because cytotoxic effect of materials is a very diversely discussed issue. In fact, many studies on cell-surface interaction report that toxic effects are not visible because cells are indeed proliferating in the reported studies. Besides, viability cell metabolism assays such as MTT reduction are often erroneously described as measuring cell proliferation without the use of proper controls to confirm effects on metabolism [1]. Also, in the mentioned review stating that ITO and gold are non-toxic, diverse effects of these materials on cell viability are reported with the overall conclusion that these materials are considered non-toxic.

In our study, we saw that the proliferation rate was different on the different materials. However, we did not give any statements in our manuscript if gold and ITO are toxic or not. It is very likely that cells suffer from growth alterations and possible metabolic effects which are interesting to focus on in future studies. We added a statement on page 10 lines 292-294.

2) From the studies in Section 2.2, the authors concluded that TiN and TiN nanocolumnar are suitable for the growth of neuron and glial cells. However, the authors did not compare their results with those growing neurons under the conventional methods. In this case, the authors are recommended to compare the results in the reports regarding the culturing of SH-SY5Y and U-87 MG.

We expect that the reviewer wants us to compare our cell data with experiments in which cells are cultured on conventional cell culture dishes. We did not use these dishes for two reasons: Firstly, we used gold and ITO as control since these materials are state-of-the-art-materials for neuro-electrodes – even if they exhibit toxic effects (see above). Secondly, we wanted to show that TiN and nano-TiN exhibit enhanced bioactive properties compared to conventional electrode materials such as gold and ITO. Besides, we would like to point out that the main focus of our study was to introduce an analytical and computer-based approach to investigate the bioactivity of a surface and show this on the example of neurons and glial cells on different materials.   

  1. Riss, T.L.; Moravec, R.A.; Niles, A.L.; Duellman, S.; Benink, H.A.; Worzella, T.J.; Minor, L. Cell Viability Assays. In Assay Guidance Manual; Markossian S, Sittampalam GS, G.A., Ed.; Eli Lilly & Company and the National Center for Advancing Translational Sciences, 2013; pp. 1–25.

Round 3

Reviewer 1 Report

1) Line 30 in Abstract, the phrase “..nanocolumnar TiN as a promising bioactive material candidate..” should be revised to “ nanocolumnar TiN as a potential bioactive material”

2) Line 106 on p.3, The biomaterials performance cannot be evaluated by “merely” measuring cell positions from fluorescent images. This phase should be revised.

Author Response

1) Line 30 in Abstract, the phrase “..nanocolumnar TiN as a promising bioactive material candidate..” should be revised to “ nanocolumnar TiN as a potential bioactive material”

We changed the word “promising” by “potential” as suggested (line 30, page 1)

2) Line 106 on p.3, The biomaterials performance cannot be evaluated by “merely” measuring cell positions from fluorescent images. This phase should be revised.

We changed “merely” by “simply” in line 106 (page 3)